# Hypermethylation at 45S *rDNA* promoter in cancers

**Trang Thi Quynh Tran[1,2]**, **Trang Hien Do[1]**, **Tung The Pham[1]**, **Phương Thi Thu Luu[1]**, **Oanh Minh Pham[1]**, **Uyen Quynh Nguyen[2]**, **Linh Dieu Vuong[3]**, **Quang Ngoc Nguyen[3]**, **Tuan Van Mai[3]**, **Son Van Ho[4]**, **Than Thi Nguyen[4]**, **Lan Thi Thuong Vo**[1,2]*

1 Faculty of Biology, VNU University of Science, Vietnam National University, Hanoi, Vietnam, 2 VNU Institute of Microbiology and Biotechnology, 3 Vietnam National Cancer Hospital, 4 Department of Chemistry, 175 Hospital, Ho Chi Minh City, Vietnam

☯ These authors contributed equally to this work.
* vothithuonglan@hus.edu.vn

**Data Availability Statement:** All relevant data are within the paper and its Supporting information files.

## Abstract

The ribosomal genes (*rDNA* genes) encode 47S rRNA which accounts for up to 80% of all cellular RNA. At any given time, no more than 50% of *rDNA* genes are actively transcribed, and the other half is silent by forming heterochromatin structures through DNA methylation. In cancer cells, upregulation of ribosome biogenesis has been recognized as a hallmark feature, thus, the reduced methylation of *rDNA* promoter has been thought to support conformational changes of chromatin accessibility and the subsequent increase in *rDNA* transcription. However, an increase in the heterochromatin state through *rDNA* hypermethylation can be a protective mechanism teetering on the brink of a threshold where cancer cells rarely successfully proliferate. Hence, clarifying hypo- or hypermethylation of *rDNA* will unravel its additional cellular functions, including organization of genome architecture and regulation of gene expression, in response to growth signaling, cellular stressors, and carcinogenesis. Using the bisulfite-based quantitative real-time methylation-specific PCR (qMSP) method after ensuring unbiased amplification and complete bisulfite conversion of the minuscule DNA amount of 1 ng, we established that the *rDNA* promoter was significantly hypermethylated in 107 breast, 65 lung, and 135 colon tumour tissue samples (46.81%, 51.02% and 96.60%, respectively) as compared with their corresponding adjacent normal samples (26.84%, 38.26% and 77.52%, respectively; $p < 0.0001$). An excessive DNA input of 1 µg resulted in double-stranded rDNA remaining unconverted even after bisulfite conversion, hence the dramatic drop in the single-stranded DNA that strictly required for bisulfite conversion, and leading to an underestimation of *rDNA* promoter methylation, in other words, a faulty hypomethylation status of the rDNA promoter. Our results are in line with the hypothesis that an increase in *rDNA* methylation is a natural pathway protecting *rDNA* repeats that are extremely sensitive to DNA damage in cancer cells.

**Funding:** We also confirm that this study was funded by Vingroup Innovation Foundation (VINIF) under project code VINIF.2022.DA00036. The funders had no role in the study design, data collection and analysis, or preparation of the manuscript, however, decided on the journal for publication.

**Competing interests:** The authors have declared that no competing interests exist.

## Introduction

The massive synthesis of ribosomal RNA that accounts for up to 80% of all cellular RNA takes place in the nucleolus. Aside from the critical function in ribosome synthesis, nucleoli also play a role in several cellular functions, including the organization of genome architecture and regulation of gene expression, in response to growth signaling, cellular stressors, and carcinogenesis [1–3].

The ribosomal genes (*rDNA* genes) are organized into large blocks, termed Nucleolar Organiser Regions (NORs), with around 400 rDNA repeats divided among the five pairs of acrocentric chromosomes (13, 14, 15, 21, and 22) in the human genome [1, 4]. Each *rDNA* gene includes the 47S rRNA encoding sequence and a non-transcribed intergenic spacer (IGS). The rRNA biogenesis is a multi-step process initiated in the nucleolus with the production of precursor 47S rRNA transcripts that are rapidly processed into the 18S, 5.8S and 28S rRNA molecules [5]. Throughout the cell cycle, *rDNA* genes coexist in three different and dynamic chromatin states: silent, inactive, and active *rDNA* genes. The silent *rDNA* state is represented by nucleosome occupancy and a heavily methylated promoter. On the contrary, the inactive and active *rDNA* states are represented by nucleosome depletion, as well as a slightly methylated or unmethylated promoter, respectively corresponding to non-transcribed or transcribed *rDNA* [6–9]. The inactive state can be turned active and *vice versa* depending on the upstream binding factor UBF whose binding is impacted by DNA methylation [7]. Switching to an active *rDNA* state co-occurs with a reorganization of *rDNA*-genome contacts and changes in gene expression [10]. Hypomethylation of the silent *rDNA* by inactivating the *DNMT1* and *DNMT3b* genes encoding DNA methyltransferases would transform them into actively transcribed *rDNA*, however, surprisingly, can result in a reduction of rRNA synthesis, nucleolar fragmentation, *rDNA* array instability and ectopic expression of cryptic rRNA genes [11, 12]. On the other hand, *de novo* methylation of *rDNA* with a nuclease-dead Cas9 (dCas9)-DNA methyltransferase fusion enzyme had no detectable effect on rRNA transcription and growth rate of human cells [13]. Thus, DNA methylation goes beyond regulating *rDNA* transcription and into maintaining *rDNA* stability and genome integrity, and controlling gene expression hub through reformed heterochromatin architecture [1, 14, 15]. Thus, profiling *rDNA* methylation, which plays a pivotal role in maintaining the balance between the open and closed chromatin states of the nucleolus implies its potential self-defense of the cell against diseases including cancer [16, 17].

In vigorously and uncontrollably growing cancer cells, the inevitable upregulation of ribosome biogenesis has been recognized as a hallmark feature [17]. Prevalent hypomethylation of *rDNA* promoter was detected in numerous types of cancer [18–20], supporting likely conformational changes of chromatin accessibility and subsequent increase in *rDNA* transcription [3, 21]. Interestingly, unchanged methylation or even hypermethylation at *rDNA* promoter was also revealed in prostate, breast, lung and oral squamous cancers [22–25], supporting the idea of a protective mechanism in which heterochromatin formation at *rDNA* repeats is required against failures due to repeat instability [23]. The inconsistent methylation status observed in cancers of the *rDNA* promoter could be explained by a varied progression of the disease stage [21] or challenged by incomplete bisulfite conversion, the exclusive approach to the quantitative measurement of *rDNA* methylation level. Indeed, incomplete bisulfite conversion has been reported in studies in which *rDNA* was either hyper- or hypomethylated [19, 22, 23, 25]. In these studies, due to incomplete bisulfite conversions, only clonal-sequencing of bisulfite-treated DNA with all cytosine residues in non-CpG dinucleotides that had been converted to thymine were included in the methylation analysis of *rDNA* promoters. The number of selected clones depended on bisulfite conversion efficiency, and thus, may have had a likely

impact on the accuracy of the methylation level obtained. An excessive DNA amount or copy number variants have been mentioned as causes of errors in DNA methylation measurement when using both bisulfite-based methods or bisulfite-free methods such as methylated DNA immunoprecipitation (MeDIP) and methyl-CpG binding domain-based capture (MBDCap) [26, 27]. Reasonably, any variance in *rDNA* methylation level, from hypo- to hyper- and even an unchanged methylation state in cancers can be influenced, at least in part, by an uncontrolled amount of input DNA (10 ng—2000 ng) subjected to bisulfite conversion [19, 22, 23, 25].

In this study, using a minuscule DNA amount of 1 ng for bisulfite conversion, we analyzed the methylation profile of the *rDNA* promoter in 307 matched pairs of tumour tissues and adjacent normal tissues from primary lung, colon and breast cancer samples. We established that the *rDNA* promoter is significantly hypermethylated in breast, lung, and colon tumour tissue samples (46.81%, 51.02% and 96.60%, respectively) as compared with their adjacent normal samples (26.84%, 38.26% and 77.52%, respectively) (p < 0.0001). An excessive DNA input amount of 1 μg and a minuscule DNA amount of 1 ng, converted by bisulfite, resulted in opposing *rDNA* promoter methylation profiles, respectively of hypomethylation and hypermethylation. Our results support the speculation that an increase in *rDNA* methylation may be a natural pathway in protecting *rDNA* repeats that are extremely sensitive to DNA damage in cancer cells [3, 28–30].

## Materials and methods

### Sample collection, DNA isolation and bisulfite conversion

Primary tumour tissue samples and their corresponding adjacent normal tissue samples were collected from fresh-frozen biopsies of 107 breast cancer patients, 135 colon cancer patients and 65 lung cancer patients. Sample classifications were done by pathologists at the Pathology and Molecular Biology Departments in National Hospitals. The recruitment period spanned from April 1st, 2023 to June 30th, 2024. Informed consent was obtained from patients in written form, and the study was approved by the Ethics Committee of the Vietnam Academy of Science and Technology (01-2023/NCHG-HDDD). DNA was extracted from fresh tissues using the DNeasy™ Blood & Tissue Kit (Qiagen) and quantified using the Qubit4™ Fluorometer (Thermo Fisher Scientific). DNA samples were subjected to bisulfite conversion using the EZ DNA Methylation-Gold™ kit (Zymo Research). Elution was performed using 20 μL of elution buffer. Two μL of bisulfite-treated DNA was used as template for real-time PCR.

### Primer design

The primer sets targeted a core promoter of the *rDNA* genes (U13369.1). The methylation-dependent-specific PCR (MSP) primers used for profiling *rDNA* methylation were derived from the CpGs-containing sequence of the core promoter [31]. The specificity of the MSP primers, annealing only to the bisulfite-treated *rDNA* sequences, was tested using (i) bisulfite-treated and (ii) untreated human genomic DNA (Promega) as templates for qPCR. Amplification products were only obtained from the reactions with bisulfite-treated DNA, ensuring the accuracy of the MSP primers designed for methylated *rDNA* targets. Primer sequences, amplicon lengths and qPCR conditions are presented in S1 Table.

### Determination of PCR amplification bias and PCR amplification efficiency

One ng of control samples with defined methylation levels (0%—100%), created by mixing human methylated DNA and human unmethylated DNA (Zymo Research), was bisulfite-

treated and used as templates for the assessment of PCR amplification biases when using either the methylation-independent PCR (MIP) primer set, specific to the reference sequence, or the MSP primer set, specific to the methylated *rDNA* sequence. PCR amplification bias was measured by either (i) plotting the CT values from amplifying the bisulfite-converted samples using the MIP primers, against the methylation level of each control sample or (ii) plotting the DNA methylation level recovered after the qPCR reactions against the input DNA methylation level of each control sample [32]. The PCR amplification efficiency was measured by plotting the CT values against a serial dilution of methylated human DNA (Zymo Research) treated by bisulfite.

## Methylation-specific quantitative PCR

*rDNA* methylation profiles were quantified via real-time PCR with a volume of 20 μL per reaction, using bisulfite-converted DNA as template and the PowerUp™ SYBR™ Green Master Mix (Thermo Fisher Scientific). Real-time PCR assays were duplexed for both of the following reactions: (i) using the MIP primers to quantify bisulfite-converted reference sequences, and (ii) using the MSP primers to quantify methylated *rDNA* sequences [33]. Water with no DNA template was included in each PCR reaction as a control for contamination.

## Methylation level calculation

Human methylated DNA (Zymo Research), defining the 100% methylation level, was used as the calibrator for the relative quantification of *rDNA* methylation level in samples [33]. Two duplicated reactions were carried out for each sample: one amplifying all bisulfite-converted reference sequences to normalise the qMSP data for each sample, and the other using the MSP primer set to quantify the methylated *rDNA*. Relative quantification of methylated *rDNA* was calculated for each sample following the Pfaffl formula [33].

## Statistical analysis

In all graphs, the *rDNA* methylation levels are represented as median values. All comparisons between more than two groups based on the quantitative values were assessed via the Kruskal-Wallis test. Comparisons between two groups based on the quantitative values were assessed via the Mann-Whitney U test for independent samples and the Wilcoxon matched-pair signed-rank test for pair-matched samples. The linear relationships of the genomic DNA input, the *rDNA* methylation level, and the CT values were evaluated through simple linear regression analysis. For all statistical analyses, a p-value of $< 0.05$ was considered significant. All analyses and graphing were performed using the GraphPad Prism program version 9 (https://www.graphpad.com/scientific-software/prism/).

# Results

## Assessment of PCR amplification bias and PCR amplification efficiency

Unequal amplification of methylated and unmethylated templates can result in inaccurate measurements of DNA methylation level, termed PCR amplification bias [32]. We investigated whether the designed MIP and MSP primer sets would produce PCR amplification bias that may impact the quantitative measurement of *rDNA* methylation level. 100% methylated and 100% unmethylated human DNA (Zymo Research) were mixed to create DNA control samples with methylation levels ranging from 0% to 100%. One ng of each control sample was bisulfite-treated and subjected as templates for assessment of PCR amplification bias using the MIP primer set specific to the reference sequence, and the MSP primer set

specific to the methylated *rDNA* sequence. Standard curves obtained by plotting the CT values against the methylation level for the MSP primer set (rDNA Me) were linear, and that for the MIP primer set (Ref) remained unchanged regardless of the change in methylation level from 0% to 100% of the templates, indicating consistent amplification of both the methylated and unmethylated templates by the MIP primers (Fig 1A). On the other hand, a serial dilution of human methylated DNA amount ranging from 5 pg to 1000 pg was bisulfite treated and used as templates in qPCR. The CT values have a strong linear relationship with the DNA input for both primer sets ($R^2 > 0.99$); however, there was a difference in amplification efficiency (E value) with 97.37% for the reference amplicons and 73.32% for the *rDNA* amplicons (Fig 1B). The difference in CT values of both the reference and the methylated *rDNA* sequences was then plotted against the logarithm of the template input amount, with the slope of the fitted line to be -0.75, thus demonstrating the Pfaffl formula suitable for the relative quantification of methylated *rDNA* (Fig 1C). Using the Pfaffl formula, the recovered *rDNA* methylation levels of the control samples were identical to the input methylation levels (Fig 1D). Taken together, these results indicate the designed MIP and MSP primer sets do not produce PCR amplification bias, thus ensuring accurate measurement of *rDNA* methylation levels using the Pfaffl formula [33].

## Identifying the impact of the DNA input amount on *rDNA* methylation assessments

As mentioned previously, an excessive input of genomic DNA, corresponding to an excessive copy number of repeated sequences used for bisulfite conversion can result in errors in DNA methylation measurement [26]. To further understand the faulty estimation of *rDNA* methylation, two different DNA quantities of human genomic DNA (Promega), 1 ng and 1000 ng, were converted by bisulfite and used as templates for qPCR with the primer sets specific to the methylated *rDNA* sequence and the reference sequence. Theoretically, the rDNA methylation level obtained should be constant regardless of the DNA amount used for bisulfite conversions. However, the *rDNA* methylation level detected from the 1000 ng input DNA was dramatically reduced (16.13%) as compared with that from the 1 ng input DNA (41.26%) (p = 0.0006) (Fig 2A). A higher DNA input amount correlated with a lower *rDNA* methylation level, indicating a faulty underestimation of *rDNA* methylation, in other words, a faulty hypomethylation status of *rDNA* sequences.

The faulty underestimation of *rDNA* methylation was confirmed by using four different DNA quantities: 1 ng, 5 ng, 50 ng, and 1000 ng of fully methylated human DNA (Zymo Research), converted by bisulfite and used as templates for qMSP with the MIP and MSP primer sets. All DNA sequences were fully methylated in this DNA, thus, PCR amplification was unbiased, and theoretically, the *rDNA* methylation level should be 100% regardless of the DNA input amount for bisulfite conversion. However, only the 1 ng and the 5 ng inputs had the correct estimation of 100% *rDNA* methylation, the 50 ng input severely overestimated *rDNA* methylation (134.63%) and the 1000 ng input severely underestimated *rDNA* methylation (52.08%) (Fig 2B). Incomplete conversion likely occurred with the 50 ng input, yielding single-stranded DNA with an excess of unconverted cytosines, thus leading to an overestimation of the methylation content. On the contrary, unconverted DNA from the 1000 ng input even after bisulfite treatment dramatically reduces single-stranded DNA, thus leading to an underestimation of the methylation level. A higher input amount of the human methylated DNA correlated with a lower *rDNA* methylation level, indicating a faulty hypomethylation status of this repeated sequence.

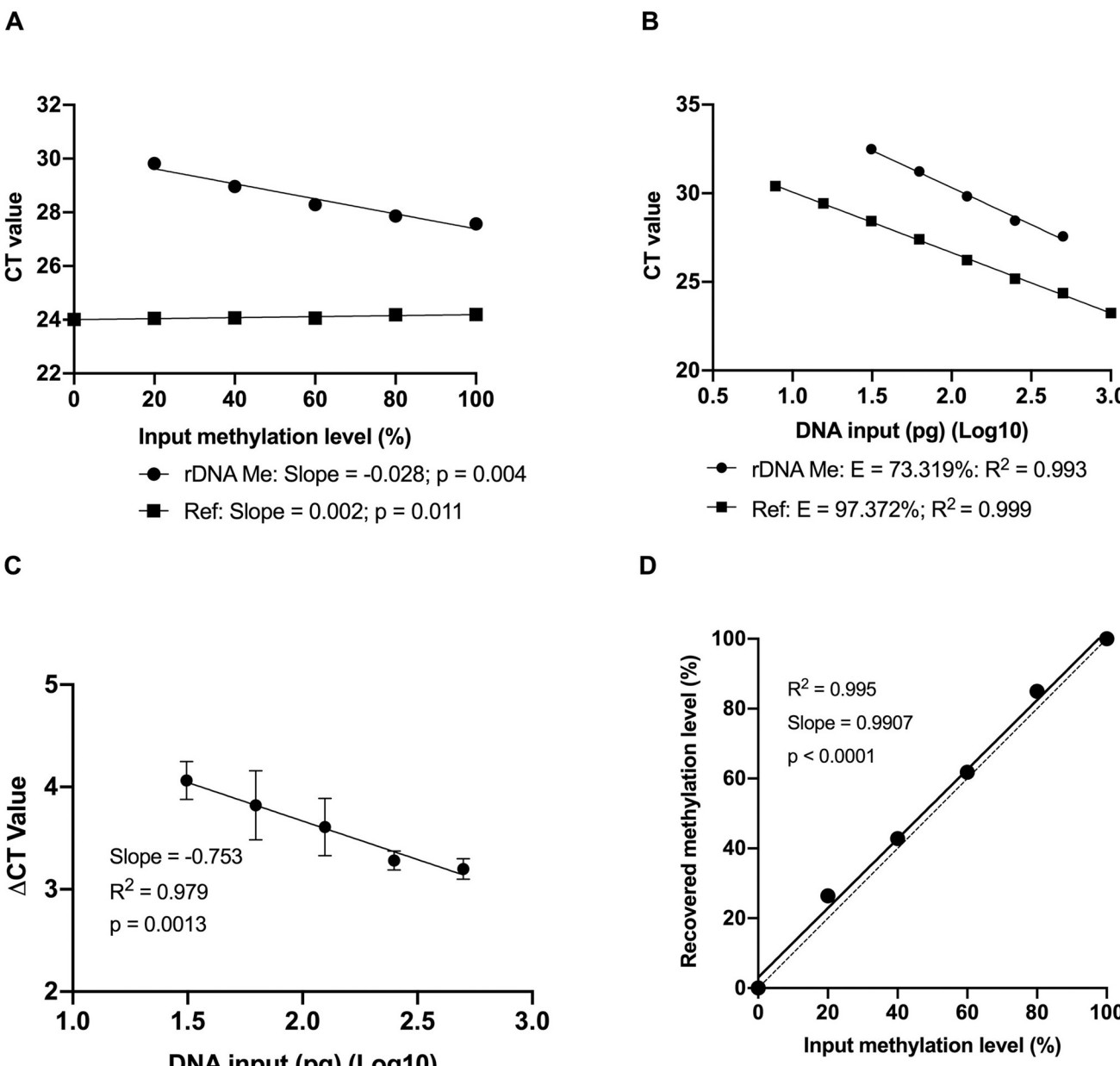

**Fig 1. Evaluation of PCR amplification bias and PCR amplification efficiency.** (**A**) Control samples with defined methylation levels ranging from 0% to 100% were used as templates in qPCR. Standard curves obtained by plotting the CT values against the methylation level for the MSP primer set (rDNA Me) were linear, and that for the MIP primer set (Ref) remained constant throughout. (**B**) A serial dilution of human methylated DNA ranging from 5 pg to 1000 pg was bisulfite-treated and used as the template for qPCR. The CT values have a strong linear relationship with the DNA input for both primer sets ($R^2 > 0.99$); however, the amplification efficiency (E value) of the reference amplicon strongly differs from that of the rDNA amplicon (24% difference). (**C**) The ΔCt value, calculated by subtracting the CT value of the MSP primer set from that of the MIP primer set for each sample, proportionally increases as the DNA input decreases, indicating the Pfaffl formula to be suitable for relative quantification. (**D**) Control samples with defined methylation levels ranging from 0% to 100% were used as templates in qPCR. Using the Pfaffl formula, recovered rDNA methylation levels were identical to the input methylation levels, demonstrating the ability to accurately identify rDNA methylation levels using the designed primer sets. Simple linear regression was used in statistical analysis. Number of observations for each assay was $\geq 3$.

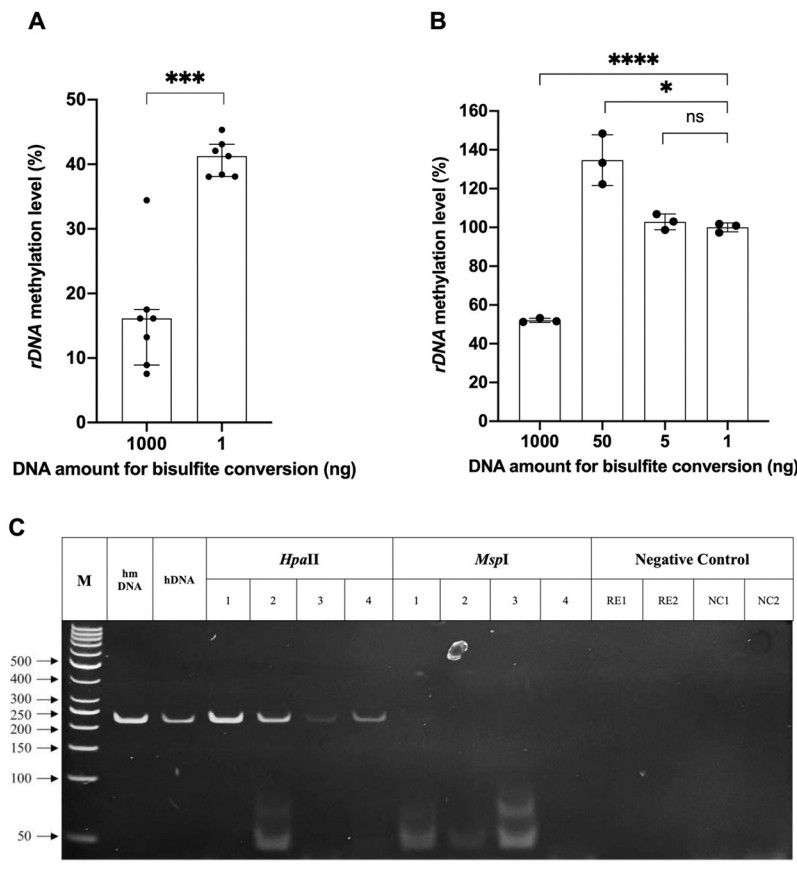

**Fig 2. An excessive DNA input resulted in enormous variation in *rDNA* methylation level and remaining DNA unconverted.** *rDNA* methylation levels enormously varied between different DNA input amounts of human genomic DNA (hDNA) (1000 ng and 1 ng) (**A**) and fully methylated human DNA (hmDNA) (1000 ng, 50 ng, 5 ng, and 1 ng) (**B**) for bisulfite conversion. (**C**) PCR products were amplified with the primers specific to the native *rDNA* sequences. The DNA amount of 1 μg from hmDNA and hDNA was treated with bisulfite and used as templates for PCR. The DNA amount of 1 μg from hmDNA (1) and hDNA (2) were bisulfite treated, digested with restriction enzymes and used as templates for PCR. The DNA amount of 100 pg from hDNA (3) and hmDNA (4) were digested with restriction enzymes and used as templates for positive control. Restriction enzyme reactions (RE1-*Hpa*II, RE2-*Msp*I) and PCR reactions without DNA (NC1, NC2) were used as templates for negative control. The Mann-Whitney U test (A) and the unpaired t-test (B) were used in statistical analysis. (ns) nonsignificant; (*) p < 0.05; (***) p < 0.001; (****) p < 0.0001.

The faulty hypomethylation of *rDNA* was likely due to insufficient yield of single-stranded DNA that is strictly required for bisulfite conversion [34]. To understand if double-stranded DNA remained after bisulfite reaction, 1 μg of human genomic DNA (hDNA) and human methylated DNA (hmDNA) were bisulfite-treated and digested by isoschizomers *Hpa*II and *Msp*I, both recognizing the sequence CCGG. *Hpa*II is blocked by methylation of either cytosine, whereas *Msp*I is blocked only by methylation of the outer cytosine. The digested products were subjected to PCR with primers specific to native *rDNA* sequences (Fig 2B). PCR products were successfully obtained only from the bisulfite-treated and *Hpa*II digested template, indicating that methylated double-stranded *rDNA* remained even after bisulfite reaction. Hence, this result supports the idea of inefficient denaturation of excessive double-stranded DNA, resulting in a decrease in *rDNA* methylation level.

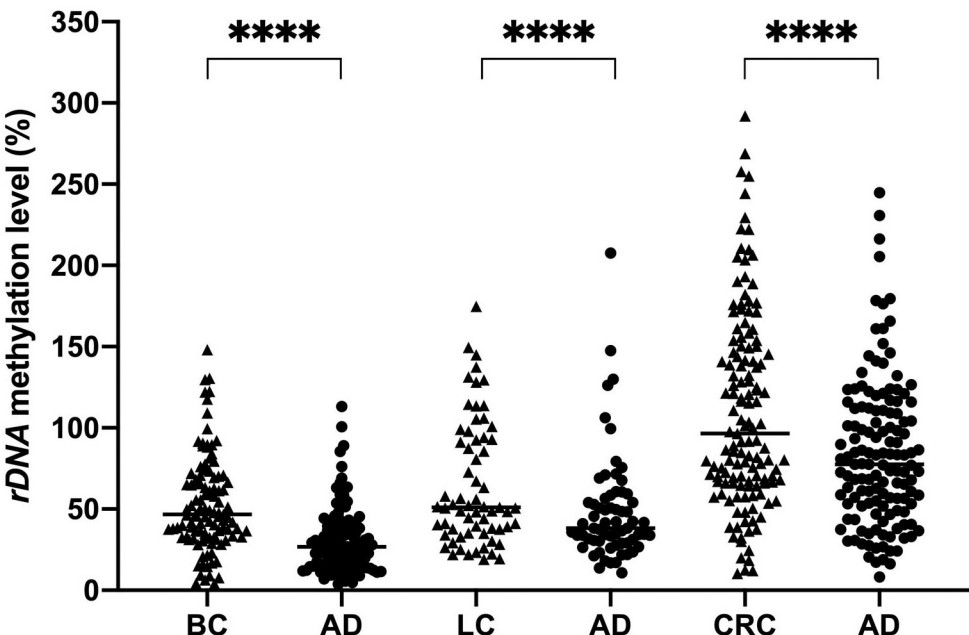

**Fig 3. *rDNA* methylation level in cancer.** *rDNA* methylation profile in tumour samples of breast cancer (BC), lung cancer (LC) and colorectal cancer (CRC) as compared with their corresponding adjacent tissue samples (AD). Methylation assessments were performed on 1 ng of DNA converted by bisulfite. The Wilcoxon matched-pairs signed-rank test was used in statistical analysis. (****) p < 0.0001.

## Analysis of *rDNA* methylation levels in tumour and pair-matched adjacent tissues

Using 1 ng of genomic DNA for bisulfite conversion, we extended the *rDNA* methylation analysis to 107 pairs of breast cancer samples (including 107 breast tumour samples and 107 corresponding adjacent tissue samples), 135 pairs of colorectal cancer samples and 65 pairs of lung cancer samples. Significant *rDNA* hypermethylation was revealed in tumour samples of breast cancer (46.81%), colon cancer (96.60%) and lung cancer (51.02%) as compared to the corresponding adjacent tissue samples (26.84%, 77.52% and 38.26%, respectively) (p < 0.0001) (Fig 3).

The association between rDNA methylation profiles and the clinicopathological features of the breast, lung and colon cancer tissues is presented in Table 1. No correlation between rDNA methylation levels and age, sex, or histologic tumour type was observed. In breast cancer, rDNA methylation levels progressively increased as tumour grades rose (p = 0.0227); this phenomenon was not observed in lung cancer or colorectal cancer. In addition, a higher *rDNA* methylation level in lung cancer correlated with the pathological stages I+II and the N0 nodal status when respectively compared with the pathological stages III+IV (p = 0.0105) and the N1 nodal status (p = 0.0013). On the other hand, rDNA methylation levels of colorectal tumours in later stages (pT4) were significantly higher than those in earlier stages (pT1-3) (p = 0.0109). Interestingly, tumours from the rectal site also had a lower methylation level when compared to the colon site (p = 0.0142).

## Discussion

Cells respond to metabolic, proliferative or differentiating states by modulating either transcription levels or epigenetic states of *rDNA* genes [35–37]. Interestingly, no more than 50% of

**Table 1. Association of *rDNA* methylation status with the clinicopathological characteristics of breast, lung and colon cancer tissues.**

| | Breast cancer (107) | | | Lung cancer (65) | | | Colorectal cancer (135) | | |
|---|---|---|---|---|---|---|---|---|---|
| | Total | Median | p-value | Total | Median | p-value | Total | Median | p-value |
| **Age** | | | | | | | | | |
| <50 | 34 | 47.828 | 0.9284[b] | 9 | 48.900 | 0.5186[b] | 14 | 121.662 | 0.4857[b] |
| ≥50 | 73 | 46.812 | | 56 | 51.246 | | 121 | 91.566 | |
| **NI** | - | | | - | | | - | | |
| **Gender** | | | | | | | | | |
| Male | - | | | 43 | 48.953 | 0.3042[b] | 85 | 91.846 | 0.9222[b] |
| Female | 107 | 46.812 | | 22 | 54.487 | | 50 | 97.175 | |
| **NI** | - | | | - | | | - | | |
| **Pathological stage** | | | | | | | | | |
| I | 11 | 45.587 | 0.8183[a] | - | | **0.0105[b]** | - | | |
| II | 19 | 49.969 | | - | | | - | | |
| III | 8 | 51.311 | | - | | | - | | |
| I+II | - | | | 44 | 54.487 | | - | | |
| III+IV | - | | | 21 | 35.184 | | - | | |
| **NI** | 69 | | | - | | | 135 | | |
| **Tumour size (Breast cancer)/Tumour stage (Colorectal cancer)** | | | | | | | | | |
| pT1 | 20 | 45.637 | 0.3815[a] | - | | | - | | **0.0109[a]** |
| pT2 | 37 | 49.276 | | - | | | - | | |
| pT1 + pT2 | - | | | | | | 31 | 87.614 | |
| pT3 | - | | | - | | | 53 | 78.234 | |
| pT4 | - | | | - | | | 19 | 132.090 | |
| pT3 + pT4 | 7 | 58.035 | | - | | | - | | |
| **NI** | 43 | | | 65 | | | 32 | | |
| **Tumour grade** | | | | | | | | | |
| I | 4 | 30.447 | **0.0227[a]** | - | | 0.1159[b] | - | | 0.7819[b] |
| II | 63 | 45.587 | | 13 | 63.437 | | - | | |
| I+II | - | - | | - | | | 115 | 92.479 | |
| III | 14 | 63.909 | | 14 | 38.093 | | 8 | 80.479 | |
| **NI** | 26 | | | 38 | | | 12 | | |
| **Nodal status** | | | | | | | | | |
| N0 | 37 | 42.530 | 0.1913[a] | 43 | 57.944 | **0.0013[b]** | - | | |
| N1 | 19 | 55.655 | | 22 | 37.795 | | - | | |
| N2 + N3 | 8 | 48.046 | | - | | | - | | |
| **NI** | 43 | | | - | | | 135 | | |
| **Histologic tumour type** | | | | | | | | | |
| IDC | 78 | 47.409 | 0.2593[b] | - | | 0.4375[b] | - | | |
| Adenocarcinoma | - | | | 57 | 51.024 | | 108 | 113.102 | |
| Other | 29 | 45.019 | | 8 | 45.604 | | 1 | 149.259 | |
| **NI** | - | | | - | | | 26 | | |
| **Sampling location (Colorectal cancer)** | | | | | | | | | |
| Colon site | - | | | - | | | 82 | 115.817 | **0.0142[b]** |
| Rectal site | - | | | - | | | 53 | 75.707 | |
| **NI** | - | | | - | | | - | | |

* a: Using the Kruskal-Wallis test

* b: Using the Mann-Whitney U test

NI: No information

*rDNA* genes are actively transcribed at any given time, and about half of *rDNA* genes are silent and form heterochromatin structures with epigenetic marks such as H3K9me2 and DNA methylation [6, 38]. Silent *rDNA* repeats are required for the efficient assembly of DNA repair factors on the highly transcribed *rDNA* genes that are hot spots of DNA double-stranded breaks (DSBs) [39]. Damaged *rDNA* repeats, predominantly repaired through homology-directed repair (HR) [40–42] are kept in the heterochromatic state, thereby reducing the mobility of the broken ends and restricting faulty recombination such as inter-chromosomal recombination [39, 43, 44]. Reactivating a large fraction of normally silent *rRNA* genes in human cells, by depleting CpG methylation through inactivation of DNMT1 and DNMT3b or treatment with DNA methylation inhibitors (aza-dC), resulted in the reduction, instead of induction, of *rRNA* synthesis, rearrangement of *rDNA* arrays and ectopic *rDNA* transcription by Pol II [11]. On the other hand, transcriptional increase in active *rDNA* genes was paradoxically induced by treatment with the Pol I transcription inhibitor [45]. Particularly, *rDNA* segments that are enriched in long-range *rDNA*-genomic interactions, are in the heterochromatin-repressed state and are associated with different cancers [28, 29]. These results demonstrated that silencing *rDNA* repeats, at least in part through DNA methylation, goes beyond transcriptional regulation and into the maintenance of *rDNA* stability and genome integrity, and regulation of gene expression hub [1, 3, 29, 30].

In cancer, due to the repetitive nature of hyperactivated transcription and rich GC sequences, making *rDNA* highly susceptible to replicative stress and replication fork stalling, *rDNA* repeats are extremely sensitive to DNA double-stranded breaks (DSBs) [39]. Consequently, damaged *rDNA* repeats are transcriptionally suppressed [46, 47] or even prohibited in the entire nucleolus if the *rDNA* is severely damaged [48]. Hence, it is plausible that *rDNA* break repairs assisted by an increase in the heterochromatin state through *rDNA* hypermethylation is a protective mechanism teetering on the brink of a threshold where cancer cells rarely proliferate successfully [49, 50]. It should be kept in mind that Pol I hyperactivity results from activation by oncogenes or a release from repression by tumour suppressors [51, 52]. Moreover, the over-expression of most genes that encode for the basal components of the Pol I transcriptional machinery is consistently found in a wide array of cancer types [53, 54]. Reasonably, the increase in ribosome production, a well-known hallmark of cancer [17] by Pol I hyperactivity, can be regulated in many alternative ways, thus does not necessarily correlate with the *rDNA* copy number [55, 56] nor the hypomethylation of *rDNA* promoter [12].

Our study investigated *rDNA* promoter methylation in solid cancers using a bisulfite-based method, and carefully addressed the challenges of incomplete bisulfite conversion and inconsistent amplification of methylated and unmethylated templates, all of which may result in inaccurate measurement of *rDNA* methylation level. We demonstrated that an excessive input DNA of 1 μg, treated by bisulfite, has a causal impact on the faulty hypomethylation status of *rDNA* sequences. Both the remaining unconverted DNA input due to excessive DNA shown in our study (Fig 2), and the renaturation during the conversion process that results in incomplete conversions, have been mentioned in bisulfite reaction protocols and commercial kit's instructions [34, 57]. A minuscule input DNA of 1 ng was appropriate to accurately quantify the methylation of *rDNA* repeats by the bisulfite-based qMSP method in which the MIP and MSP primer sets showed no PCR amplification bias (Fig 1). Based on well-founded evidence, hypermethylation of *rDNA* promoter was revealed in tumour tissues from breast, colon and lung cancer samples when compared to their corresponding adjacent normal tissue samples (Fig 3). This result is in line with previous studies in which *rDNA* hypermethylation was determined by Southern blot analysis [24, 58, 59], and supports an important role of *rDNA* methylation, especially of its promoter, to be one of the key factors in carcinogenesis [16, 60, 61].

The limitation of this study is the focus on only the methylation of the *rDNA* promoter, instead of studying the methylation levels of the gene body, including the sequences encoding 18S, 5.8S and 28S RNAs, which exhibit differently during normal development, ageing, diseases and cancer [62]. Moreover, *rDNA* copy number and its relation with *rDNA* methylation should be necessarily further investigated. The number of *rDNA* repeats varies between individuals and is unstable in different cancer types [55, 56]. On the other hand, *rDNA* methylation status is related to the clinicopathological characteristics in breast, colon and lung cancer tissues (Table 1), therefore, investigating the *rDNA* methylation in circulating-cell-free DNA (cfDNA), which retains the epigenetic characteristics of the tissue from which it was released [63], should provide a promising tool for improving markers of cancer screening, accurate disease-progression surveillance and improvement of treatment [20, 21].

## Supporting information

**S1 Table. Primer sets and quantitative real-time PCR conditions for measurement of *rDNA* methylation and detection of the native *rDNA* sequences.**
(DOCX)

**S1 Dataset.**
(XLSX)

## Author Contributions

**Data curation:** Trang Thi Quynh Tran, Trang Hien Do, Tung The Pham, Oanh Minh Pham.

**Formal analysis:** Trang Thi Quynh Tran, Trang Hien Do, Tung The Pham, Phưởng Thi Thu Luu, Oanh Minh Pham, Than Thi Nguyen.

**Funding acquisition:** Lan Thi Thuong Vo.

**Methodology:** Lan Thi Thuong Vo.

**Project administration:** Lan Thi Thuong Vo.

**Resources:** Linh Dieu Vuong, Quang Ngoc Nguyen, Tuan Van Mai, Son Van Ho.

**Validation:** Trang Thi Quynh Tran, Tung The Pham.

**Visualization:** Trang Thi Quynh Tran, Trang Hien Do, Tung The Pham, Phưởng Thi Thu Luu.

**Writing – original draft:** Trang Thi Quynh Tran, Trang Hien Do, Tung The Pham, Uyen Quynh Nguyen.

**Writing – review & editing:** Uyen Quynh Nguyen, Lan Thi Thuong Vo.

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
