## [Decision Letter · Decision Letter 0]

6 Sep 2024

PONE-D-24-35450Hypermethylation at 45S rDNA promoter in cancersPLOS ONE

Dear Dr. Vo,

Thank you for submitting your manuscript to PLOS ONE. After careful consideration, we feel that it has merit but does not fully meet PLOS ONE’s publication criteria as it currently stands. Therefore, we invite you to submit a revised version of the manuscript that addresses the points raised during the review process, 

We look forward to receiving your revised manuscript.

Kind regards,

Abdul Rauf Shakoori

Academic Editor

PLOS ONE

**Journal Requirements:**

This study was funded by the Vingroup Innovation Foundation (VINIF) under project code VINIF.2022.DA00036.

Reviewers' comments:

Reviewer's Responses to Questions

**Comments to the Author**

1. Is the manuscript technically sound, and do the data support the conclusions?

Reviewer #1: Yes

Reviewer #2: Partly

2. Has the statistical analysis been performed appropriately and rigorously? 

Reviewer #1: Yes

Reviewer #2: No

3. Have the authors made all data underlying the findings in their manuscript fully available?

Reviewer #1: Yes

Reviewer #2: Yes

4. Is the manuscript presented in an intelligible fashion and written in standard English?

Reviewer #1: Yes

Reviewer #2: Yes

5. Review Comments to the Author

**Reviewer #1:** Trang et al. presented nice findings on "Hypermethylation at 45S rDNA promoter in cancers". I congratulate authors for their effort in presenting hypermethylation status of 45S rDNA in cancers partcularly on Lung, colon and Breast.

I would suggest authors to extend this study on others cancer too to find out whether qunatity of initial rDNA does have any association with methylation status.

**Reviewer #2: **1. «An excessive DNA input of 1 µg resulted in an increased unconverted double-stranded rDNA, thus leading to an underestimation of rDNA promoter methylation, in other words, a faulty hypomethylation status of the rDNA promoter»

Could you clarify how this point can be interpreted?

Incomplete conversion during bisulfite treatment (including incomplete denaturation of double-stranded DNA) leads to an excess of unconverted cytosines. That is, we overestimate the methylation content. If the methylation-specific PCR was balanced, then the underconverted 1000 ng should have received the hypermethylated status.

The hypomethylated status with incomplete conversion of 1000 ng could have been obtained with predominant amplification of unmethylated DNA, which is inconsistent with the statement about balanced methylation-specific PCR. In this case, it is recommended to use an endogenous control for the sample - a DNA region with a given methylation level. For example, this could be an imprinted region where the ratio of methylated and unmethylated alleles is 50/50

2. «One ng of control samples with defined methylation levels (0 % - 100 %)…»

What method was used to normalize the samples, since the methylation level in some cases exceeds 100%?

3. How would you rate rRNA expression, rDNA hydroxymethylation status and proteins responsible for active demethylation (particularly TETs), as well as the involvement of transcription factors and methylated cytosine binding factors (UBF)?

6. PLOS authors have the option to publish the peer review history of their article (what does this mean?). If published, this will include your full peer review and any attached files.

Reviewer #1: **Yes: **Niyaz A Naykoo

Reviewer #2: No

---

## [Author Response · Author response to Decision Letter 0]

9 Sep 2024

Vo Thi Thuong Lan, Assoc. Prof.

Faculty of Biology, VNU University of Science,

334, Nguyen Trai, Thanh Xuan

Ha Noi, Viet Nam.

 PLoS ONE Editorial Board

Ha Noi, September 09th, 2024 

Dear Members of the Editorial Board,

We are deeply grateful to the Reviewers for the supportive comments and suggestions that have allowed us to considerably improve our manuscript titled “Hypermethylation at 45S rDNA promoter in cancers” by Tran et al., submitted to PLoS ONE under the reference PONE-D-24-35450. We have now rewritten the manuscript after meticulous consideration of all the comments from the Reviewers and highlighted the changes in the manuscript, which are detailed point by point in the following section. We have added a panel in Figure 2 that describes the hypomethylation of the rDNA promoter using different amounts of fully methylated human DNA for bisulfite conversion.

We confirm that this study was funded by Vingroup Innovation Foundation (VINIF) under project code VINIF.2022.DA00036. The funders had no role in the study design, data collection and analysis, or preparation of the manuscript, however, decided on the journal for publication.

Reviewer #1: I would suggest authors to extend this study on others cancer too to find out whether quantity of initial rDNA does have any association with methylation status.

We are grateful to the Reviewer for this comment. We are currently investigating rDNA methylation levels in gynecologic cancer and gastrointestinal cancer as well as in circulating cell-free DNA (cfDNA) from cancerous patients and healthy individuals. We are also interested in the association, if any, between rDNA copy number and its methylation in cancer.

Reviewer #2:

1. «An excessive DNA input of 1 µg resulted in an increased unconverted double-stranded rDNA, thus leading to an underestimation of rDNA promoter methylation, in other words, a faulty hypomethylation status of the rDNA promoter». Could you clarify how this point can be interpreted?

Our results in Fig. 2 demonstrated that an excessive DNA input of 1 µg led to an underestimation of rDNA promoter methylation (Fig. 2A), which resulted from double-stranded rDNA remaining unconverted in the form of native DNA even after bisulfite treatment (Fig. 2B).

This underestimation can be interpreted by a decrease in single-stranded rDNA templates occurring in bisulfite reactions due to the reannealing of the excessive DNA input. It should be noted that bisulfite conversions occur strictly on single-stranded DNA, and an increase in the input double-stranded rDNA copy number could favourably promote renaturation, thus resulting in a decrease in the required single-stranded rDNA. This phenomenon is similar to that in PCR reactions in which product yield would dramatically decline when the double-stranded DNA template is over the threshold, thus causing renaturation and reducing the single-stranded templates. It is worth noting that an input of 1 µg of genomic DNA for methylation analysis – the amount recommended by most manufacturers – is equivalent to 106 copies of single-locus targets, with only 2 copies per diploid genome, however, would be equivalent to 109 copies of the rDNA gene, with up to around 400-500 copies per genome. Our previous study has established that a copy number higher than 108 of repeated sequences can lead to incomplete bisulfite conversion (Ref. 26). In this study, we have additionally clarified that the error in bisulfite conversion is due to the DNA input remaining unconverted as native double-stranded DNA (Fig. 2B). Thus, to present this finding more clearly, the statement should be adjusted to “An excessive DNA input of 1 µg resulted in double-stranded rDNA remaining unconverted even after bisulfite conversion, hence the dramatic drop in the single-stranded DNA that strictly required for bisulfite conversion, and leading to an underestimation of rDNA promoter methylation, in other words, a faulty hypomethylation status of the rDNA promoter”. We have adjusted this statement in the rewritten manuscript (page 2).

Incomplete conversion during bisulfite treatment (including incomplete denaturation of double-stranded DNA) leads to an excess of unconverted cytosines. That is, we overestimate the methylation content. If the methylation-specific PCR was balanced, then the underconverted 1000 ng should have received the hypermethylated status.

We fully agree with the Reviewer on incomplete bisulfite conversions, including incomplete denaturation of double-stranded DNA, yielding single-stranded DNA containing an excess of unmethylated cytosines, thus, overestimating the methylation content. However, our study has demonstrated that instead of incomplete conversion, the double-stranded DNA input remained unconverted even after bisulfite treatment in the form of native double-stranded DNA (Fig. 2B), hence the dramatic drop in the single-stranded DNA that strictly required for bisulfite conversion, and leading to an underestimation of methylation levels. Our first result (Fig. 1) has already established the balance of our methylation-specific PCR reactions, thus, it is the under-converted 1000 ng of DNA input that led to the hypomethylation status, instead of hypermethylation.

The hypomethylated status with incomplete conversion of 1000 ng could have been obtained with predominant amplification of unmethylated DNA, which is inconsistent with the statement about balanced methylation-specific PCR. In this case, it is recommended to use an endogenous control for the sample - a DNA region with a given methylation level. For example, this could be an imprinted region where the ratio of methylated and unmethylated alleles is 50/50.

We fully agree with the Reviewer on the predominant amplification of unmethylated DNA impacting the methylation level. Having understood the detriment of unequal PCR amplification, we have first assessed PCR bias and confirmed the equal amplification of both the MIP and the MSP primer set (Fig. 1). In addition, as recommended by the Reviewer, we have used fully methylated (100 %) human DNA (Zymo Research), in place of an imprinted region where the ratio of methylated and unmethylated alleles is 50/50, for direct assessment of the rDNA methylation status. Four different DNA quantities: 1 ng, 5 ng, 50 ng and 1000 ng of fully methylated human DNA were converted by bisulfite and used as templates for qMSP with the MIP and MSP primer sets. All DNA sequences were fully methylated in this DNA, thus, PCR amplification was unbiased, and theoretically, the rDNA methylation level should be 100 % regardless of the DNA input amount for bisulfite conversion. However, only the 1 ng and the 5 ng inputs had the correct estimation of 100 % rDNA methylation, the 50 ng input severely overestimated rDNA methylation (134.63 %) and the 1000 ng input severely underestimated rDNA methylation (52.08 %). Incomplete conversion likely occurred with the 50 ng input, yielding single-stranded DNA with an excess of unconverted cytosines, thus leading to an overestimation of the methylation content. On the contrary, unconverted DNA from the 1000 ng input even after bisulfite treatment dramatically reduces single-stranded DNA, thus leading to an underestimation of the methylation level. A higher input amount of the human methylated DNA correlated with a lower rDNA methylation level, indicating a faulty hypomethylation status of this repeated sequence. We have adjusted this result in Fig. 2 of the rewritten manuscript (Fig. 2B, page 10-11).

2. «One ng of control samples with defined methylation levels (0 % - 100 %) …»

What method was used to normalize the samples, since the methylation level in some cases exceeds 100 %?

We thank the Reviewer for having considered the method used to normalize the samples. As described in Materials and Methods, the rDNA methylation level is calculated by using the ΔΔCT method, the comparative CT method commonly used for quantitative methylation-specific PCR (qMSP) [1, Ref. 33]. The ΔΔCT method requires a calibrator sample with a known methylation level of the target specifically recognized by the MSP primer set and normalized through a reference specifically recognized by the MIP primer set. Moreover, the ∆∆CT method is only valid when the amplification efficiencies of the target and reference sequences are similar. If the amplification efficiencies of the two amplicons are not the same, an alternative formula, the Pfaffl formula must be used to determine the relative quantification of the methylated target in different samples (Ref. 33).

Our study used 1 ng of fully methylated human DNA, treated by bisulfite, as the calibrator with a known rDNA methylation level of 100 %. On the contrary, 1 ng of control samples with defined methylation levels (0 % - 100 %), created by mixing human methylated DNA and human unmethylated DNA, was bisulfite-treated and used as templates for the assessment of PCR amplification biases.

As the Reviewer remarked, the methylation level in some cases exceeded 100%. This is most likely the result of an increase in rDNA copy number, which may be unstable in cancer (Ref. 16, 55, 56). Alternatively, rDNA getting increasingly fragmented due to its sensitivity to DNA double-stranded breaks may also be a cause. In addition, this phenomenon also seems to be dependent on the type of cancer, as colorectal cancer is shown to have several cases with methylation levels exceeding 100 %, yet this number is rather scarce for breast cancer. Thus, the copy number of the full-length and the truncated rDNA genes should be carefully quantified through the copy number of promoter as well as the sequences encoding 18S, 5.8S and 28S RNAs in further analysis.

3. How would you rate rRNA expression, rDNA hydroxymethylation status and proteins responsible for active demethylation (particularly TETs), as well as the involvement of transcription factors and methylated cytosine binding factors (UBF)?

We are truly interested in the Reviewer’s questions. Based on our best knowledge, Pol I specifically and solely act on the rDNA promoter, producing the precursor 47S rRNA transcripts that are then processed into the 18S, 5.8S and 28S rRNA transcripts, whose amounts vary in cancer (Ref. 22). Thus, rRNA expression assessments should carefully quantify a ratio among these transcripts.

Two main transcription factors, the selectivity factor SL1 and the upstream binding transcription factor (UBF), comprise the pre-initiation complex (PIC) and initiate rRNA transcription. The presence of CpG methylation at the rDNA promoter inhibits UBF binding, thus abrogating the formation of the PIC complex (Ref. 7). However, UBF depletion, leading to an increase in the proportion of silent rDNA genes, has minimal effects on overall cellular rDNA transcription [2, Ref. 7], which is consistent with the idea that rDNA transcription can be regulated in many alternative ways, and does not necessarily correlate with the hypomethylation of rDNA promoter.

The TET enzymes catalyze the methyl group of 5-methylcytosine (5mC) in DNA to generate 5-hydroxymethylcytosine (5hmC), 5-formylcytosine (5fC) and 5-carboxylcytosine (5caC). 5hmC levels, expression and activity of TET enzymes are reduced in various human cancers [3-5]. Interestingly, TET deficiency can lead to the re-localization of the DNMT3 enzymes that mediate de novo DNA methylation from the heterochromatin region to the euchromatin region [6,7]. Thus, it is reasonable to speculate that through DNA methylation, the active rDNA state can be switched into the inactive state, which is required for repairing DNA double-stranded breaks (DSBs) on the rDNA genes (Ref. 39).

References

1. Livak KJ, Schmittgen TD. Analysis of relative gene expression data using real-time quantitative PCR and the 2(-Delta Delta C(T)) method. Methods. 2001;25(4): 402-408. doi: 10.1006/meth.2001.1262 PMID: 11846609

2. Theophanous A, Christodoulou A, Mattheou C, Sibai DS, Moss T, Santama N. Transcription factor UBF depletion in mouse cells results in downregulation of both downstream and upstream elements of the rRNA transcription network. J Biol Chem. 2023; 299(10): 105203. doi: 10.1016/j.jbc.2023.105203 PMID: 37660911

3. Salmerón-Bárcenas EG, Zacapala-Gómez AE, Torres-Rojas FI, Antonio-Véjar V, Ávila-López PA, Baños-Hernández CJ, et al. TET Enzymes and 5hmC Levels in Carcinogenesis and Progression of Breast Cancer: Potential Therapeutic Targets. Int. J. Mol. Sci. 2024, 25(1):272. Doi: 10.3390/ ijms25010272 PMID: 38203443

4. Alrehaili AA, Gharib AF, Alghamdi SA, Alhazmi A, Al-Shehri SS, Hagag HM, et al. Evaluation of TET Family Gene Expression and 5-Hydroxymethylcytosine as Potential Epigenetic Markers in Non-small Cell Lung Cancer. In Vivo. 2023 Jan-Feb;37(1):445-453. doi: 10.21873/invivo.13098 PMID: 36593050

5. Jeschke J, Collignon E, Fuks F. Portraits of TET-mediated DNA hydroxymethylation in cancer. Curr Opin Genet Dev. 2016; 36: 16-26. doi: 10.1016/j.gde.2016.01.004 PMID: 26875115

6. López-Moyado IF, Tsagaratou A, Yuita H, Seo H, Delatte B, et al. 2019. Paradoxical association of TET loss of function with genome-wide DNA hypomethylation. PNAS. 2019; 116(34):16933-16942. doi: 10.1073/pnas.1903059116 PMID: 31371502

7. Charlton J, Jung EJ, Mattei AL, Bailly N, Liao J, et al. TETs compete with DNMT3 activity in pluripotent cells at thousands of methylated somatic enhancers. Nat. Genet. 2020; 52(8):819-827. doi: 10.1038/s41588-020-0639-9 PMID: 32514123.

---

## [Editor Report · Decision Letter 1]

12 Sep 2024

Hypermethylation at 45S rDNA promoter in cancers

PONE-D-24-35450R1

Dear Dr. Vo,

We’re pleased to inform you that your manuscript has been judged scientifically suitable for publication and will be formally accepted for publication once it meets all outstanding technical requirements.

Kind regards,

Abdul Rauf Shakoori

Academic Editor

PLOS ONE
---

## [Editor Report · Acceptance letter]

4 Nov 2024

PONE-D-24-35450R1 

PLOS ONE

Dear Dr. Vo, 

I'm pleased to inform you that your manuscript has been deemed suitable for publication in PLOS ONE. Congratulations! Your manuscript is now being handed over to our production team.

Kind regards, 

on behalf of

Prof. Dr. Abdul Rauf Shakoori 

Academic Editor

PLOS ONE